# An Alternative to PCA for Estimating Dominant Patterns of Climate Variability and Extremes, with Application to U.S. and China Seasonal Rainfall

**Stephen Jewson** 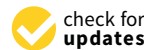

Risk Management Solutions Ltd., Peninsular House, 30 Monument Street, London EC3R 8NB, UK; stephen.jewson@gmail.com; Tel.: +44-(0)7858-393370

**Abstract:** Floods and droughts are driven, in part, by spatial patterns of extreme rainfall. Heat waves are driven by spatial patterns of extreme temperature. It is therefore of interest to design statistical methodologies that allow the rapid identification of likely patterns of extreme rain or temperature from observed historical data. The standard work-horse for the rapid identification of patterns of climate variability in historical data is Principal Component Analysis (PCA) and its variants. But PCA optimizes for variance not spatial extremes, and so there is no particular reason why the first PCA spatial pattern should identify, or even approximate, the types of patterns that may drive floods, droughts or heatwaves, even if the linear assumptions underlying PCA are correct. We present an alternative pattern identification algorithm that makes the same linear assumptions as PCA, but which can be used to explicitly optimize for spatial extremes. We call the method Directional Component Analysis (DCA), since it involves introducing a preferred direction, or metric, such as "sum of all points in the spatial field". We compare the first PCA and DCA spatial patterns for U.S. and China winter and summer rainfall anomalies, using the sum metric for the definition of DCA in order to focus on total rainfall anomaly over the domain. In three out of four of the examples the first DCA spatial pattern is more uniform over a wide area than the first PCA spatial pattern and as a result is more obviously relevant to large-scale flooding or drought. Also, in all cases the definitions of PCA and DCA result in the first PCA spatial pattern having the larger explained variance of the two patterns, while the first DCA spatial pattern, when scaled appropriately, has a higher likelihood and greater total rainfall anomaly, and indeed is the pattern with the highest total rainfall anomaly for a given likelihood. The first DCA spatial pattern is arguably the best answer to the question: what single spatial pattern is most likely to drive large total rainfall anomalies in the future? It is also simpler to calculate than PCA. In combination PCA and DCA patterns yield more insight into rainfall variability and extremes than either pattern on its own.

**Keywords:** principal component analysis; PCA; directional component analysis; DCA; empirical orthogonal functions; Empirical Orthogonal Function (EOF); extremes; U.S. rainfall; China rainfall

## 1. Introduction

Principal Component Analysis (PCA), also known as Empirical Orthogonal Function (EOF) analysis, is often used in climate research and related fields for analysing correlated data in two or more dimensions. PCA is widely used because of its mathematical elegance, mathematical properties and simplicity. It has various applications, such as filling gaps in historical datasets [1] and rapid identification of patterns of variability [2]. When being used to tackle some problems, however, limitations of PCA may become apparent, and this has led to the development of various extensions of PCA, each addressing a different issue. For instance, the spatial patterns identified by PCA tend to fill

the spatial domain being analysed, and in some cases it would be more appropriate to identify more localised patterns. This led to the development of rotated EOF analysis, as studied in, for instance, Mestas-Nunez [3] and Lian and Chen [4], and used in Chen and Sun [5]. In another extension, known as extended EOF analysis, PCA has been used to understand developments of patterns in time [6], and in yet another has been adapted to better handle skewed data [7]. PCA and related methods have been discussed in text books such as Wilks [8], von Storch and Zwiers [9] and Jolliffe [10] and in the review paper Hannachi et al. [11].

In a recent project to develop methods to improve the resilience of financial institutions to drought shocks by identifying the patterns of rainfall that might drive the largest droughts over the domain being analysed [12] we have become aware of a property of PCA, that, in the context of the goals of this particular project, is a shortcoming. When applied to observed rainfall anomalies, the first PCA pattern maximises explained variance, by definition, but the spatial pattern does not necessarily maximise the total rainfall anomaly in any sense. The total rainfall anomaly in the first PCA spatial pattern could even be zero or very close to zero. This is simply a result of the mathematical definition of PCA and what PCA is designed to capture. As a result, the first PCA spatial pattern may not be particularly relevant in terms of its impact via floods (or droughts) on the scale of the spatial domain being analysed since such phenomena are likely to be, at least to some extent, related to the size of the total anomaly over the spatial domain. One could imagine that other spatial patterns, selected based on a total rainfall anomaly criterion of some sort, may be at least as relevant, and possibly more relevant.

Motivated by this observation, this article has the goal of deriving and testing a statistical methodology that allows us to identify the most likely spatial pattern of extreme rain, or extreme temperature. In other words, to find the pattern that answers the question: which single spatial pattern is most likely to drive large-scale rainfall or temperature anomalies in the future? To this end, we have studied a new pattern identification scheme that we call DCA (Directional Component Analysis). This scheme seeks to identify patterns which are both likely to occur, based on an analysis of the historical data (so that they are relevant), and that contain a large total rainfall or temperature anomaly (so that they have a large impact). Mathematically, to find such a pattern, we will not define the first spatial pattern as the unit length spatial pattern which maximises explained variance, as PCA does, but rather we will define the first spatial pattern as the unit length spatial pattern which has the highest likelihood for a given level of total rainfall anomaly. This spatial pattern is also, conversely, the unit length spatial pattern with the greatest rainfall anomaly for a given likelihood of occurrence. PCA and DCA spatial patterns can be scaled (i.e., multiplied by a single constant value across the whole spatial pattern) to create new spatial patterns, and when appropriately scaled, the first DCA spatial pattern *both* has a higher likelihood *and* contains a greater total rainfall or temperature anomaly than the first PCA spatial pattern, and as such would indeed be expected to be more relevant for understanding extremes such as floods, droughts or heatwaves. In this article we will restrict most of our discussion to the properties of the first spatial patterns that can be derived using PCA and DCA, as opposed to the subsequent spatial patterns and all related time-series. The spatial patterns we derive are considered as possible realisations of spatial patterns that may occur in the future.

In Section 2, we give an overview of PCA, as a basis for comparison with DCA. We give two derivations for PCA: the first, based on explained variance, is the more common. The second, based on maximising likelihood, is less usual but is useful here because it makes a link to DCA. We also apply PCA to an example dataset consisting of U.S. winter rainfall anomalies. In Section 3, we give two derivations of DCA. The first is similar to the likelihood derivation for PCA, while the second is based on regression. We then apply DCA to the same example dataset. In Section 4, we discuss the PCA and DCA results from the example dataset in detail. We also analyse the scaling properties of the DCA results in this section, since understanding the scaling properties is important for a complete understanding of the mathematical properties of DCA. In Section 5, we present results for three more examples, based on U.S. summer rainfall and China winter and summer rainfall. Together, the results show clearly that the first DCA spatial pattern yields additional insights when studying possible spatial

extremes, above and beyond what the first PCA spatial pattern can tell us. In Section 6, we summarize and conclude. In the supporting information, we give two simple examples of PCA and DCA to illustrate the DCA method. The first (Section S1) is a numerical example with a $2 \times 2$ covariance matrix. The second (Section S2) gives the general solution for PCA and DCA for any diagonal $2 \times 2$ covariance matrix. In Sections S3–S8, we prove the orthogonality of the first two DCA patterns and provide some proofs of the main optimality properties of DCA.

## 2. Principal Component Analysis

We now review PCA as a basis for comparison with DCA. We will focus on the application of PCA and DCA to rainfall anomalies, although they can both also be applied to other variables. Spatial patterns of variability of rainfall anomalies have various mathematical properties, including explained variance (defined below), length (based on considering patterns as vectors), total rainfall anomaly and likelihood (which is the same as probability density, and which can be calculated in the context of a statistical model such as the multivariate normal distribution). PCA can be described as an attempt to answer the question: what single spatial pattern best represents the variability in the data? PCA is a mathematical method that considers two of the four properties listed above (explained variance and length) and answers that question by finding the spatial patterns and time-series pairs with the greatest explained variance, among all spatial patterns of the same length. The definition of PCA quite deliberately does not take account of the total rainfall anomaly across the spatial pattern and so would not be expected to give a good representation of spatial extremes. The *standard* definition of PCA also does not take likelihood into account, although it can be reformulated in terms of likelihood (see Section 2.2 below), since likelihood and explained variance are closely related.

### 2.1. PCA Standard Derivation

We now give the standard mathematical derivation of PCA. Consider a space-time dataset of anomalies $X$ with spatial dimension $s$ and temporal dimension $t$. In the example shown below and discussed in detail in Section 4, and in the further examples shown in Section 5, we will use gridded maps of U.S. and China winter and summer rainfall anomalies for 114 years from CRU (Harris et al. [13]). Harris et al. [13] took individual station observations and interpolated them onto a 0.5 degree grid, and we use this interpolated data. We calculate anomalies from the 114 year seasonally varying time mean and then create winter and summer averages. The time dimension $t$ is either 113 (for winter) or 114 (for summer) and the space dimension $s$ is either 3319 (for the U.S.) or 3798 (for China).

We will consider mathematical methods in which we look for a spatial pattern (or patterns) that capture certain properties of the space-time variability in the data $X$. We will write an as-yet-undetermined spatial pattern as $g$, and we will vary $g$ to try and achieve the properties desired.

If the data $X$ (in our U.S. winter rainfall example, a 3319 by 113 matrix) is projected onto an unknown $s \times 1$ spatial pattern vector $g$ (the pattern we wish to solve for) the $t \times 1$ time series $p$ of amplitudes of the projection is given by the vector-matrix product:

$$p^T = g^T X \tag{1}$$

This is the time series of amplitudes that corresponds to the pattern $g$, in the data $X$ (in our U.S. winter rainfall example, this time series is a vector of 113 annual values). The variance $v$ of this time series $p$ is a scalar and is given by:

$$v = \frac{1}{t} p^T p = \frac{1}{t} g^T X X^T g = g^T C g \tag{2}$$

where $C = \frac{1}{t} X X^T$ is the $s \times s$ empirical covariance matrix of the data $X$. $C$ can be easily calculated from the data (in our U.S. winter rainfall example it is a 3319 by 3319 matrix). The scalar $v$ is known as the explained variance of the pattern $g$ in the dataset $X$. The explained variance is large either if the

pattern $g$ contains large values, or if the pattern $g$ is similar in shape to patterns that have occurred frequently in the historical data $X$, since those shapes are captured by the covariance matrix $C$.

We can imagine varying the vector $g$, subject to the constraint that $g$ is unit length (i.e., that $g^T g = 1$), and trying to maximise the variance $v$. The unit length constraint is required because otherwise the variance $v$ increases without limit as the length of $g$ increases. Using the constraint allows us to fix the length and vary the shape and structure of the pattern only.

Mathematically, we can find the maximum of a function, while applying a constraint, using the method of Lagrange multipliers. In our case, this works by maximising the Lagrange function $c$ defined as:

$$c = v - \lambda(g^T g - 1) = g^T C g - \lambda(g^T g - 1) \tag{3}$$

where $c$ is a scalar cost function, $v$ is the quantity being maximised and $\lambda$ is a scalar known as the Lagrange multiplier that multiplies $(g^T g - 1)$, the expression that defines the constraint. In this equation $g^T C g$ is largest for long vectors that project highly onto the eigenvectors of $C$ (i.e., are consistent with the historical variability in $X$, as captured in $C$). The $g^T g - 1$ term constrains length, but does not influence direction of the solution.

Differentiating with respect to the vector $g$ to find the maximum of $c$ gives

$$\frac{dc}{dg} = 2Cg - 2\lambda g \tag{4}$$

Setting equal to zero gives the equation:

$$Cg = \lambda g \tag{5}$$

This is an eigenvector equation, the solutions of which are the eigenvectors of $C$, also known as the left singular vectors, or EOFs, of $X$. These eigenvectors can be interpreted as spatial patterns that might occur in the future. The first eigenvector has the property that it maximises the explained variance in the data $X$, the second that it maximises the explained variance in what is left after the first eigenvector has been removed, and so on. The eigenvectors form an orthonormal set, and the time series associated with the eigenvectors also form an orthonormal set.

We note that there are various alternative ways to define PCA, with different terminology. For instance, finding the first PCA pattern can be defined (entirely equivalently to the above derivation) as finding a linear combination of the time series from all the spatial points such that the linear combination maximises variance while the weights within the linear combination (known as the loadings) satisfy a length constraint. The spatial pattern then consists of the loadings.

The first PCA pattern for U.S. winter rainfall anomalies from the CRU data is given in Figure 1, and discussed in more detail in Section 4 below. We calculate the first PCA spatial pattern by calculating the anomaly covariance matrix of the winter rainfall anomalies and applying the solution given above.

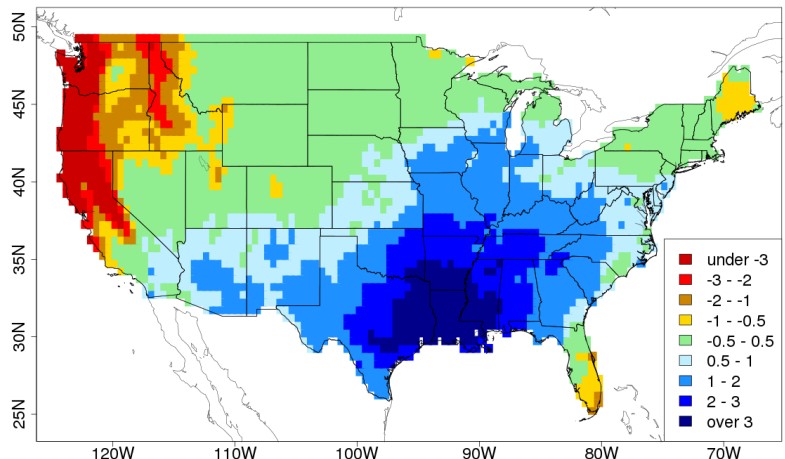

**Figure 1.** The first PCA spatial pattern for U.S. winter rainfall anomalies. This pattern maximises explained variance.

### 2.2. PCA Alternative Derivation

We now give an alternative derivation of PCA. This derivation allows us to make a link between PCA and the main derivation of DCA given below. We will see that the DCA derivation is almost the same, but with a slightly modified constraint.

### 2.2.1. Use of the Multivariate Normal Distribution

We will now assume that each spatial pattern in the historical dataset set *X* (that is: each column of *X*, each corresponding to a fixed time) is a single realisation of a multivariate random variable (also known as a vector random variable) from a multivariate normal distribution. Multivariate normal distributions are specified by two parameters: the mean and the covariance matrix. The mean of the multivariate normal in our case is a spatial pattern consisting of only zeroes (corresponding to no rain anywhere) because we are considering anomalies. The covariance matrix of the multivariate normal is estimated from data in the standard way as given above and captures the variance of rainfall anomalies at each individual spatial point and the covariances of rainfall anomalies between spatial points. Any possible spatial pattern lies within the support of the multivariate normal, and so for each possible spatial pattern we can calculate the probability density, or likelihood, from the expression for probability density for the multivariate normal. This gives a measure of how likely that spatial pattern is to occur, within the context of the distribution. Saying we are modelling the rainfall anomalies using a multivariate normal in this way is equivalent to saying that we are modelling the rainfall at each individual spatial location as normally distributed (so that the rain at each point in time is a single realisation from the normal distribution for that location), and that we model the dependencies between the rainfall anomalies at different locations using the multivariate normal structure.

Within the fitted multivariate normal, spatial patterns with small rainfall anomalies, which are close to the mean, are more likely to occur and will have high values for the likelihood, while spatial patterns with large rainfall anomalies, which are far from the mean, are less likely to occur and will have lower values for the likelihood. The pattern which is most likely to occur, and which has the highest likelihood, is the mean itself (in the same way as would be the case for a univariate normal distribution). Also, patterns that have a spatial structure highly consistent with the covariance matrix (and hence are similar to the patterns that occur in the historical data, since the covariance matrix is derived from the historical data) are more likely to occur, and will have higher likelihoods, while patterns that have spatial structure which is less consistent with the covariance matrix (and are less similar to the patterns that occur in the historical data) are less likely to occur and will have lower

likelihoods. In the alternative derivation of PCA, we try to find the unit vector spatial pattern with the highest likelihood in the fitted multivariate normal distribution, instead of that which explains the most variance. Put simply, we are just looking for the most likely pattern. The unit vector constraint in this case simply avoids the solution collapsing to the mean, which is zero everywhere. Likelihood itself is awkward to manipulate for the multivariate normal distribution, but instead, and equivalently, we can maximise log-likelihood, which is more convenient. The log-likelihood considered as a function of the unknown spatial pattern $g$ for the multivariate normal is proportional to a mathematical quantity known as the Mahalanobis consistency, which is minus one times another mathematical quantity known as the Mahalanobis distance squared, which is given by $M^2 = g^T C^{-1} g$, where $C^{-1}$ is the inverse or pseudoinverse of the covariance matrix $C$. Mahalanobis consistency and Mahalanobis distance have simple intuitive interpretations. Mahalanobis consistency is a reasonable general measure for consistency of a spatial pattern vector $g$ with a covariance matrix $C$, and hence implicitly measures consistency of a vector $g$ with the original data $X$ i.e., consistency with what has been seen in the historical data. High values mean more consistent: spatial patterns that are similar to recurring patterns in the historical data will have a high Mahalanobis consistency. Spatial patterns that are very different to what has happened in the past will have a low Mahalanobis consistency. Mahalanobis distance is just a measure of distance from the mean, taking into account spatial covariances. It can be considered a multivariate generalisation of z values, which is the number of standard deviations from the mean in a univariate normal [8]. High values of Mahalanobis distance mean further from the mean, and hence lower likelihood, just as they would for high z values in the univariate normal distribution. The connections between PCA, the multivariate normal distribution, the Mahalanobis consistency and the Mahalanobis distance are discussed in detail in Wilks [8] and von Storch and Zwiers [9].

### 2.2.2. Derivation

For the alternative derivation of PCA we once again apply the Lagrange multiplier formulation: this time to find the pattern that maximises the log-likelihood subject to the constraint of unit length. Maximising the log-likelihood without the constraint simply gives the mean pattern, which is zero everywhere since we are considering anomalies. The Lagrange function for this new problem has two terms: one for the log-likelihood term $-M^2$, and one for the unit length constraint as before, and is given by:

$$c = -M^2 - \lambda(g^T g - 1) = -g^T C^{-1} g - \lambda(g^T g - 1) \tag{6}$$

In this equation the $-g^T C^{-1} g$ term is largest (most positive) for short vectors, and vectors that project highly onto the eigenvectors of $C^{-1}$ and $C$ (which are the same). The solutions of this equation are also the eigenvectors of $C$, and so are also the PCA spatial patterns i.e., the solution is the same as the solution to the first (and more standard) derivation of PCA given above, even though the formulation of the problem is different.

The data does not in fact have to be multivariate normal for this derivation (and the subsequent derivation of DCA) to make sense. In non-normal cases, PCA, by this derivation can be described as a method that finds the unit vector that maximises the Mahalanobis consistency, $-M^2$, rather than finding the unit vector that maximises the log-likelihood.

### 2.3. Two-Dimensional Example

PCA is illustrated in Figure 2 for a simple case. The two axes represent rainfall anomaly amounts in two locations. The ellipse represents a single contour of constant likelihood (which is also a contour of constant log-likelihood, constant Mahalanobis consistency and constant Mahalanobis distance) from the joint probability distribution of rainfall at these locations (with higher likelihoods inside the ellipse). All points on this contour are equally likely to occur. Points inside are more likely to occur, and points outside are less likely to occur. The most likely point is the origin, which is the mean of the distribution. Other contours of constant likelihood, for different levels of likelihood, would be concentric with the

contour shown, but are omitted for clarity. The principal axis of the ellipse, illustrated by the double arrow, is tilted slightly to the left of vertical, indicating a negative correlation between rainfall at these two locations. The first PCA spatial pattern is a scaled version of this principal axis vector (scaled to be a unit vector), and consists of negative rainfall anomalies (or loadings) at location one (horizontal axis), and positive rainfall anomalies (or loadings) at location two (vertical axis), reflecting the negative correlation. In this example we will assume that the likelihood value that defines the ellipse has been chosen so that the PCA arrow shown is in fact the exact unit-scaled first PCA spatial pattern. The two diagonal lines represent contour lines of constant total rainfall anomaly, summed across the two locations, and therefore the highest total rainfall anomaly amounts are in the top right hand corner of the figure. The total rainfall anomaly of the first PCA spatial pattern is not particularly large in this case since there is a cancellation of rainfall anomalies to some extent between the two locations, because of the negative correlation.

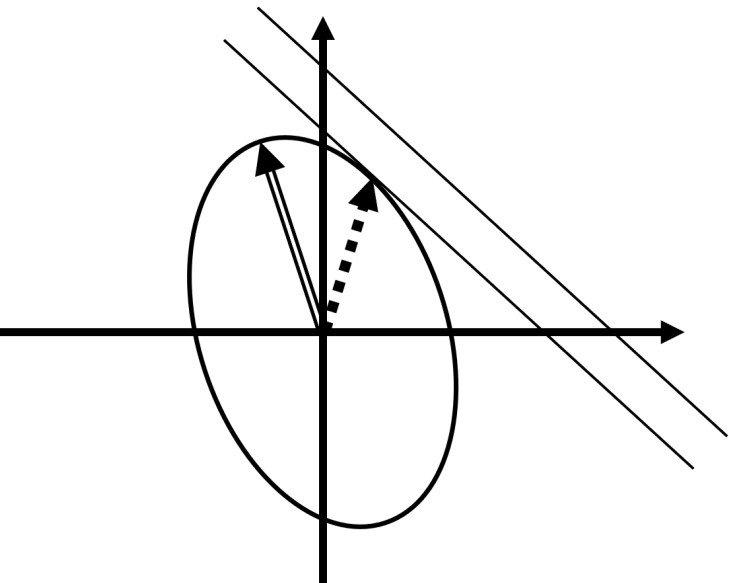

**Figure 2.** PCA and DCA spatial patterns in a space with two dimensions. The axes are the two dimensions, which might be, for instance, rainfall anomaly amounts at two locations. The diagonal lines then show lines of constant total rainfall anomaly. Assuming that the two variables are bivariate normal distributed in space with a weak negative correlation the ellipse shows a contour of constant likelihood (probability density) or constant Mahalanobis consistency, with higher likelihoods (higher Mahalanobis consistency, lower Mahalanobis distance) inside the ellipse. Each point in this two dimensional space represents a spatial pattern, consisting of a single realisation from the bivariate normal, made up of rainfall anomaly values at the two locations. The tip of the double arrow gives the rainfall anomaly values for the first PCA spatial pattern, while the tip of the dotted arrow gives rainfall anomaly values for a scaled version of the first DCA spatial pattern. In this case the two patterns are scaled to have the same likelihood, as we can see from the fact that they both just touch the ellipse. The PCA spatial pattern has larger explained variance, while the DCA spatial pattern (which is the point on the ellipse with the greatest total rainfall anomaly, by definition) captures a greater total rainfall anomaly, and so is more relevant for understanding extreme rainfall totals.

The dotted arrow is the first DCA spatial pattern and is explained in the next section.

## 3. Directional Component Analysis

Having reviewed PCA, we now describe DCA. Certain possible shortcomings of PCA have been discussed in the introduction. In particular, PCA does not maximize total rainfall anomaly in any way,

and our goal is to find a statistical method that finds a pattern that does. Put another way, whereas PCA tries to identify the single spatial pattern that best represents variability, we now try to identify the single spatial pattern that best represents extreme totals across the field.

We have seen above that PCA is based on the explained variance and the length of the spatial pattern. In the alternative derivation it is based on likelihood and length of the spatial pattern. To achieve its different goal, DCA considers a different pair of properties: likelihood and total rainfall anomaly. The definition of DCA quite deliberately does not take account of explained variance, since the goal is to understand extremes rather than variance. We see from this that PCA and DCA consider different aspects of a pattern and not surprisingly they have different uses, different possible interpretations, and give different results.

For the same space-time dataset $X$ as considered in Section 2, again considered to be multivariate normal over the spatial dimension, and a new unknown spatial pattern $g$, we derive DCA by solving for the highest likelihood spatial pattern given a certain level of total rainfall anomaly. Explained another way: many different rainfall anomaly spatial patterns can give the same total rainfall anomaly; which of them is the most likely to occur? (from the point of view of understanding possible future extremes, presumably the most likely one is the most interesting one to look at first). Conversely, but equivalently, we could say we are looking for the spatial pattern with the greatest total rainfall anomaly given a certain value of the likelihood. Explained another way: many different rainfall anomaly spatial patterns are equally likely; which of them has the highest total rainfall anomaly? (again, this is presumably the most interesting spatial pattern to look at first from the point of view of understanding extremes). Essentially we are trying to find the pattern with the greatest total rainfall anomaly, while factoring in the requirement that the pattern should have a reasonably high probability of occurring in reality, so that it is relevant. Since the actual probability of any individual pattern occurring in a continuous distribution is zero, we use probability density (likelihood) instead of probability itself. Another reason we include likelihood in the derivation is because the pattern with the greatest rainfall anomaly, not factoring in likelihood, is simply uniform rainfall everywhere, which is not an interesting result. The use of likelihood brings the structure and patterns in the historical data into the analysis. In summary, if PCA is an attempt to find the single pattern that can tell us the most about spatially correlated variability in the dataset, then DCA is an attempt to find the single pattern that can tell us the most about the extreme high or low values of the spatial total of the anomalies in the dataset, taking spatial correlations into account. The DCA method generalizes to non-normal data in the same way that the second derivation of PCA given above in Section 2.2 does: for non-normal data we can restate the problem using Mahalanobis consistency rather than likelihood by saying we are looking for the pattern that shows the highest Mahalanobis consistency with the covariance matrix, for a given level of total rainfall anomaly, or, conversely, the pattern with the greatest total rainfall anomaly, for a given level of Mahalanobis consistency.

### 3.1. Scaling

Both PCA and DCA spatial patterns can be scaled to create new spatial patterns with the same spatial structure but different amplitude. In the context of the multivariate normal, these new spatial patterns are also possible values for the random variable (i.e., are possible realisations from the distribution) but will have different likelihoods and different total rainfall anomaly amounts. Scaling any pattern by a factor greater than 1 increases the total rainfall anomaly, but moves the spatial pattern further from the mean of zero, and hence increases the Mahalanobis distance, decreases the Mahalanobis consistency and decreases the likelihood. Scaling with a very large factor would lead to such large rainfall anomalies that the pattern could never realistically occur: this would be reflected in the likelihood values, which become very low for very large anomalies. Scaling with different factors occurs when PCA and DCA spatial patterns are combined with their time series to reconstruct the original data.

The first definition of DCA involves specifying a level of total rainfall anomaly. DCA patterns derived for different given levels of total rainfall anomaly will turn out to have the same spatial patterns but different amplitudes. This is a consequence of using the multivariate normal distribution, and is related to the fact that the multivariate normal assumes that correlations do not depend on the amplitude of the pattern. Ways to move beyond this assumption are discussed in Section 6. To make the first DCA pattern uniquely defined we normalize the amplitude to unit length. This normalized pattern can then be scaled up or down to give the optimal pattern for any given value of total rainfall anomaly (or, conversely, the optimal pattern for any given value of likelihood). This will become clearer when we discuss our U.S. winter rainfall anomaly example in detail in Section 4.

### 3.2. Two Dimensional Example

A scaled version of the first DCA spatial pattern is illustrated in Figure 2 by the dotted arrow. The scaling in the diagram has been chosen so that the dotted arrow hits the same contour of likelihood as the first PCA spatial pattern, and so occurs with the same likelihood. The scaled first DCA spatial pattern points to a greater extent towards the region of greatest total rainfall anomaly in the top right hand corner of the diagram, even though it is shorter than the first PCA spatial pattern. It is this "rotation" relative to the direction of the first PCA spatial pattern which gives the first DCA spatial pattern its particular properties: by pointing towards the region of high total rainfall anomaly, but staying on the same contour, the first DCA spatial pattern achieves a greater total rainfall anomaly amount than the first PCA spatial pattern for the same likelihood, and hence we would argue is more useful for understanding extremes of total rainfall anomaly.

One can imagine scaling the lengths of the two spatial pattern arrows in Figure 2 in different ways to create patterns with different levels of total rainfall anomaly and different likelihoods. Scaling simply changes the length of the vector, but not the direction. We can use this to illustrate various properties of PCA and DCA. For instance, if we were to scale (and lengthen) the first PCA spatial pattern in Figure 2 so that it would hit the same total rainfall anomaly line as the scaled first DCA spatial pattern shown it would contain more rainfall than before but would extend outside the elliptical contour and would hence occur with a lower likelihood than the scaled DCA pattern shown. Conversely if we were to scale the DCA spatial pattern to be slightly shorter, then it can be seen that it would still achieve higher rainfall than the first PCA spatial pattern, but would have a higher likelihood because it would fall inside the elliptical contour. This latter case is the most interesting since it creates a pattern which is both more likely and has a greater rainfall anomaly than the first PCA spatial pattern and is hence more relevant to understanding spatial extremes from both the magnitude and the likelihood perspectives. These scaling illustrations show general properties of PCA and DCA that are discussed further below, proven in the supplementary materials and illustrated in the example in Section 4. Geometrically, we can summarize DCA using Figure 2 very simply: DCA allows us to find the point on the ellipse that has the greatest rainfall anomaly (i.e., is the furthest to the top right of the diagram).

### 3.3. DCA Main Derivation

We can derive an expression for the first DCA pattern as follows. The derivation follows closely the alternative derivation of PCA given above. The total rainfall anomaly in the unknown pattern $g$ that we will be solving for is given by the sum of the rainfall anomalies at the individual locations in $g$. It can be written in a general form as a linear function of the components in $g$ as the vector dot product $g^T r$, where $r$ is the vector (1, 1, 1 ..., 1), a pattern of uniform rainfall anomaly. The analysis below also applies to any other value for the vector $r$, hence the name 'directional' component analysis: the use of $r$ introduces a preferred direction, or metric, in addition to the directions defined by the eigenvectors of the covariance matrix $C$. The inclusion of a preferred direction distinguishes this method from PCA and related methods. It therefore makes most sense to consider DCA not as a variant of PCA, but as a different approach.

Once again we will use the Lagrange multiplier framework to maximise the log-likelihood while satisfying a constraint, as we did in the alternative derivation of PCA given in Section 2.2 above. The only difference is that the constraint has changed to one of total rainfall, rather than unit length.

We then maximise the log-likelihood, for a given level of total rainfall anomaly $a$, by combining $-M^2$ and $g^T r$ in the Lagrange function as follows:

$$c = -M^2 + 2\lambda(g^T r - a) = -g^T C^{-1} g + 2\lambda(g^T r - a) \tag{7}$$

We have added an arbitrary factor of 2 in the definition of $\lambda$ to simplify the algebra later. In this equation both terms are influenced by both the length and direction of $g$. Compared with the Lagrange function used in the alternative derivation of PCA in Section 2.2, only the second term is different.

Differentiating with respect to the vector $g$ gives

$$\frac{dc}{dg} = -2C^{-1}g + 2\lambda r \tag{8}$$

and setting equal to zero gives:

$$C^{-1}g = \lambda r \tag{9}$$

which gives the solution, or set of solutions:

$$g = \lambda C r \tag{10}$$

With this equation we have succeeded in finding a simple and easily calculated expression for patterns which maximise the likelihood for a given level of total rainfall anomaly.

The second derivative of $c$ is:

$$\frac{d^2 c}{dg^2} = -2C^{-1} \tag{11}$$

and this confirms that the solution is a maximum.

The solution in Equation (10) is not unique because of the scaling by $\lambda$. As discussed above, we will define the first DCA spatial pattern $g_1$ uniquely by normalizing the solution to unit length, giving:

$$g_1 = \frac{Cr}{|Cr|} = \frac{Cr}{\sqrt{r^T C^2 r}} \tag{12}$$

We see that the solution for the first DCA spatial pattern $g_1$ is very simple. When $r$ is a vector of 1's, the first DCA spatial pattern is simply proportional to the sums of rows in the covariance matrix. Calculating the first DCA spatial pattern in practice is therefore simply a matter of calculating the covariance matrix of the data, summing the rows, and normalising to unit length.

Some mathematical observations about this solution can be made as follows:

- The given level of total rainfall anomaly $a$ does not appear in the definition of $g_1$: the first DCA spatial pattern is the same whatever the level of rainfall anomaly specified, because of the normalisation to unit length. The first DCA spatial pattern is a property of the covariance matrix (and hence of the original data) only.
- Equation (10) above gives a set of solutions based on different values for $\lambda$. The different solutions are simply different scalings of the same vector $Cr$.
- Each value of $\lambda$ corresponds to a solution for a different value of $a$.
- Given any value of $\lambda$ we can calculate $g$ (using $g = \lambda Cr$), and given $g$ we can calculate $a$ (using $a = g^T r$) and $M^2$ (using $M^2 = -g^T C^{-1} g$), giving one complete solution to the constrained maximisation problem, for a given value of $a$.

- For instance, $g_1$, which corresponds to a value of $\lambda = \frac{1}{|Cr|}$, solves the constrained maximisation problem for a given value of total rainfall anomaly of $a = g_1^T r$, and the Mahalanobis consistency achieved at the maximum in that case is $-g_1^T C^{-1} g_1$.
- Other values of $\lambda$ give solutions with different values for the total rainfall anomaly and the Mahalanobis consistency: the larger the value of $\lambda$, the larger the total rainfall anomaly, the lower the Mahalanobis consistency (and the lower the likelihood).
- All the different solutions can be derived from the unit length solution $g_1$ just by rescaling.
- The extent to which the DCA pattern has achieved its goal, of finding a pattern that has a greater likelihood for the same rainfall anomaly than PCA, can be measured by comparing the ratio $a/M$ between the first PCA and DCA spatial patterns. Because $a$ and $M$ are both linear multiples of $\lambda$, the $\lambda$'s cancel and this ratio does not depend on $\lambda$. DCA would be expected to have a higher value for this ratio, and indeed DCA can also be derived based on the idea of maximising this ratio. We will evaluate this ratio for all our examples below.

We can also solve the converse definition of DCA and maximise the total rainfall anomaly for a given value of the log-likelihood $b$, using the Lagrangian:

$$c = -\lambda'(g^T C^{-1} g - b) + 2g^T r \tag{13}$$

We omit the details, but this leads to the same set of solutions, and the same definition of $g_1$ as given above.

The first DCA spatial pattern for U.S. winter rainfall is given in Figure 3, and discussed in detail in Section 4 below.

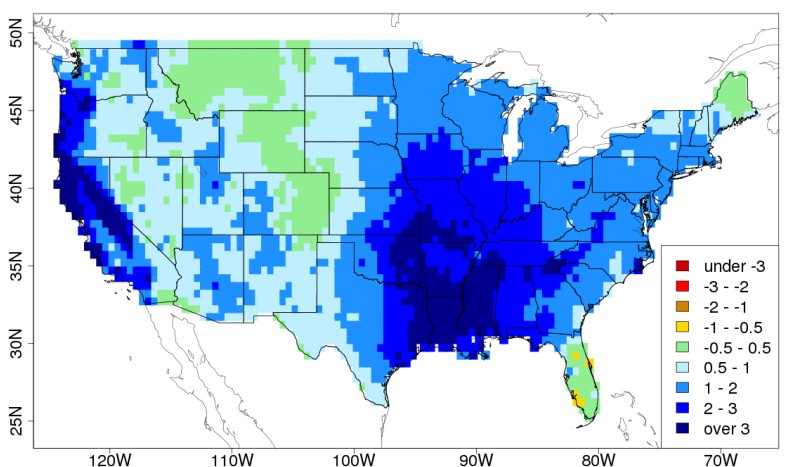

**Figure 3.** The first DCA spatial pattern for U.S. winter rainfall anomalies. With appropriate scaling this pattern is both more likely and represents a greater total rainfall anomaly than any given scaling of the first PCA spatial pattern. As a result it is more appropriate for understanding possible extremes in total rainfall anomalies than the first PCA spatial pattern.

*3.4. Derivation of the Second DCA Pattern*

In PCA different values of the Lagrange multiplier $\lambda$ correspond to the first, second, third PCA patterns, and so on. This is not the case for DCA. In DCA, as we have seen above, different values of the Lagrange multiplier simply correspond to different scalings of the same vector, and the first spatial pattern is defined as the unit length version of this vector. Second and subsequent DCA patterns have to be constructed explicitly as follows.

We can derive a time-series corresponding to the first DCA spatial pattern by projecting the original data $X$ onto the spatial pattern, giving $X^T g_1$. An approximate reconstruction of $X$ can then

be created by combining the first DCA spatial pattern with this time series, giving $g_1(g_1^T X)$, and the residuals $X_2$ from this approximation can be derived as:

$$X_2 = X - g_1(g_1^T X) \tag{14}$$

The second DCA spatial pattern can be derived by repeating the entire DCA analysis given above on this second dataset $X_2$, using $C_2 = \frac{1}{t} X_2 X_2^T$, and giving $g_2 = \frac{C_2 r}{|C_2 r|}$. This process can be continued to derive a series of spatial patterns, with corresponding time series, and as with PCA the number of spatial pattern time series pairs is equal to the rank of $X$. This series of spatial patterns will be mutually orthonormal, by construction, and so together will form an orthonormal set. The orthogonality of the first two patterns is proved in the supporting information (Section S3).

There are in fact an infinite number of possible orthonormal sets of spatial patterns, of which the PCA and DCA spatial patterns are both examples. However, there is only one orthonormal set for which the time-series of different patterns are uncorrelated, which is PCA. The time-series for different DCA patterns are correlated, except in the degenerate case when the DCA patterns are the same as the PCA patterns.

The set of DCA spatial patterns are empirical, and orthogonal, and so one could refer to them as empirical orthogonal functions (EOFs). However, the term EOF analysis is currently used synonymously with PCA, and so should perhaps reserved for that usage.

### 3.5. Regression-Based Derivation

An alternative regression-based derivation of the DCA spatial patterns proceeds as follows.

First, we construct a time series $T$ of the total rainfall anomaly at each point in time, by projecting the data $X$ onto the uniform rainfall vector $r$:

$$T = X^T r \tag{15}$$

We then regress the data $X$ onto this time series, to give regression slopes $\beta$:

$$X = \beta T^T + \epsilon \tag{16}$$

The standard estimator for $\beta$ is given by:

$$\begin{align}
\beta &= XT(T^T T)^{-1} \tag{17}\\
&= XX^T r(T^T T)^{-1} \tag{18}\\
&= tCr(T^T T)^{-1} \tag{19}\\
&\propto Cr \tag{20}
\end{align}$$

and we see that $\beta$ is proportional to the first DCA spatial pattern $g$.

Once again, the second pattern can be produced by removing the data explained by the pattern $\beta$, and repeating the process, and the series of patterns thus obtained will be the same as those derived in the previous section.

### 3.6. Properties of DCA

We now summarise some of the mathematical properties of the first DCA spatial pattern, with the assumption that $r$ is again a vector of 1's and $X$ represents rainfall anomalies, as illustration. From the derivations of the first PCA and DCA spatial patterns we can say that:

- Since PCA is designed to maximise explained variance, the explained variance of the first DCA pattern will be less than or equal to the explained variance of the first PCA pattern. The explained variance will only be equal to that of the first PCA pattern in the degenerate case that the first

PCA spatial pattern equals the vector $r$, in which case the first DCA spatial pattern will also equal $r$.

- Since the first DCA spatial pattern is designed to maximise total rainfall anomaly (for a given value of likelihood) the total rainfall anomaly $g_1^T r$ for the first DCA spatial pattern will be greater than or equal to that of the first PCA spatial pattern. In Figure 2 this corresponds to the DCA vector reaching further into the region of large total rainfall anomaly in the top right hand corner of the diagram. Once again it will only be equal in the degenerate case. This property can be shown by comparing the definition of DCA with the second definition of PCA given above. It is also proven more carefully in the supporting information, Sections S4 and S5.

- If the first DCA spatial pattern is scaled to have the same total rainfall anomaly as the first PCA spatial pattern, it will have a higher or equal likelihood, equal only in the degenerate case. This property can be shown from the definitions of PCA and DCA, and is also proven in the supporting information, Section S6.

- If the first DCA spatial pattern is scaled to have the same likelihood as the first PCA spatial pattern (which is how the arrows are scaled in Figure 2) it will have a greater or equal value for the total rainfall anomaly, equal only in the degenerate case. This property follows from the definitions, but is also proven carefully in the supporting information, Section S7.

- In the non-degenerate case, the first DCA spatial pattern can be scaled to the in-between case where it has *both* more rainfall *and* a higher likelihood than the first PCA spatial pattern. This is the most interesting property of DCA in comparison with PCA, and is the property which suggests that DCA is the better method for identifying spatial extremes (defined here as extremes in the total anomaly summed across the pattern). It is proven in the supporting information, Section S8.

We will illustrate these properties with our U.S. winter rainfall example below.

## 4. Application of DCA to Observed U.S. Winter Rainfall

We have already shown the first PCA and DCA spatial patterns for U.S. winter rainfall in Figures 1 and 3. The patterns in Figure 1 and 3 are shown as unit vectors, corresponding to the definitions. These are both patterns that could occur in reality, if the multivariate normal assumption is correct.

### 4.1. Discussion of Pattern Structure

The PCA spatial pattern in Figure 1 shows a large region of negative (red) anomalies in the north west (NW) and a large region of positive (blue) anomalies in the south east (SE), over the Mississippi basin. The DCA pattern, on the other hand, shows positive anomalies in the NW, and a larger region of positive anomalies in the east. The PCA pattern suggests that there is a recurrent pattern of climate variability on this timescale consisting of little rainfall in the NW and heavy rainfall in the SE. The DCA pattern does not contradict this suggestion. However, it does suggest in addition that there is a recurrent pattern of climate variability on this timescale that includes *both* heavy rainfall in the NW *and* the SE, with the area of heavy rainfall in the SE extending further northwards. This latter pattern is more interesting than the PCA pattern with respect to possible nationwide floods or droughts. What precise statements we can make with regards to the relative levels of rainfall and likelihood of the two patterns is explored in remainder of this section.

The DCA analysis has worked as intended. The use of DCA analysis has led to the easy and rapid identification of a possible pattern of rainfall variability that could drive large scale floods or droughts. Possible patterns of rainfall variability could also be explored by detailed analysis of the pair-wise correlations between rainfall anomalies in different regions. That would, however, be far more laborious. The DCA pattern yields rapid new insights simply by multiplying the correlation matrix with a vector.

### 4.2. Statistics of the Unit Vector Patterns

The explained variances of the PCA and DCA patterns are 18.2% and 16.8% respectively: as would be expected from the definition of PCA, the first PCA pattern has higher explained variance, although the values are very close. We can calculate the total rainfall anomaly in each of these spatial patterns by summing across the points within the spatial pattern. The PCA spatial pattern has a total rainfall anomaly of 13.2, and the DCA spatial pattern has a total rainfall anomaly of 47.9: as would be expected from the definition of DCA the DCA pattern has higher total rainfall anomaly, which in this case is higher by a factor of 3.63. The Mahalanobis distance values of the two patterns are $M_{PCA} = 2.32 \times 10^{-3}$ and $M_{DCA} = 2.47 \times 10^{-3}$, respectively. By this measure the first DCA pattern is a little further from the mean (so has a lower likelihood). The ratio of rainfall to Mahalanobis distance is 5699 for the PCA pattern and 19,407 for the DCA pattern (3.41 times higher). These ratios indicate that the DCA analysis has achieved one of its goals, which is to find a pattern with greater total rainfall anomaly for the same Mahalanobis distance (and hence the same likelihood). In this case the total rainfall anomaly in the DCA pattern is 3.41 times greater, for any given level of likelihood.

The ratio of likelihoods of the two patterns can be calculated from the Mahalanobis distance values as $pd_{DCA}/pd_{PCA} = \exp\left(M_{PCA}^2 - M_{DCA}^2\right)$, and is extremely close to 1 i.e., the patterns have almost the same likelihood. This is because both patterns are scaled to represent very small anomalies by the requirement that they are unit vectors, and because the density of a multivariate normal distribution is almost flat near the mean. If the patterns were both scaled to represent larger anomalies then the difference in likelihoods would be larger: this is explored below in Section 4.7.

To summarize these statistics we can say that in relation to the first PCA pattern the first DCA pattern has a slightly lower explained variance, a very slightly lower likelihood, and significantly higher rainfall.

### 4.3. Scaled Patterns

The most interesting quantitative properties of the first DCA spatial pattern, as compared to the first PCA spatial pattern, become apparent when we adjust the scaling of one or both of the patterns, and we will now discuss some examples of scalings in detail. The properties of the scaled patterns illustrate that the first DCA spatial pattern is more suitable than the first PCA spatial pattern for understanding possible future extremes of total rainfall anomaly. In practice, one may not need to calculate these alternative scalings, and it may be sufficient to simply consider the pattern structure, and the ratio of total rainfall anomaly to Mahalanobis distance. The ratio of total rainfall anomaly to Mahalanobis distance summarizes in one number how much the DCA pattern dominates the PCA pattern in terms of total rainfall anomaly.

We start our discussion of different possible scalings with a graphical explanation and then describe the quantitative analysis in more detail below. Figure 4 shows the total rainfall anomaly and the Mahalanobis distance value $M$ for the first PCA spatial pattern (large blue dot "P") and for the first DCA spatial pattern (large red dot "D"). That the DCA pattern has greater total rainfall anomaly but higher Mahalanobis distance (and hence lower likelihood) is shown by the fact that the DCA pattern is higher up and to the right of the PCA pattern.

If we scale the first PCA spatial pattern we generate new patterns that lie on the blue line in Figure 4. This line is straight because total rainfall anomaly and Mahalanobis distance are both linear multiples of the scaling. The slope of this line equals the ratio of total rainfall anomaly to Mahalanobis distance for the PCA pattern. This line extends from the bottom left, where patterns have rainfall anomalies very close to zero, and have low Mahalanobis distance and high likelihood, to the top right, where patterns have larger rainfall anomalies, greater Mahalanobis distance and lower likelihood.

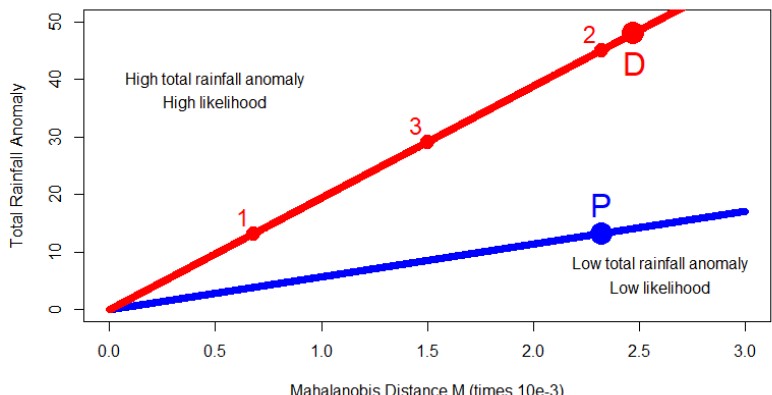

**Figure 4.** The total rainfall anomaly and Mahalanobis distances for the PCA and DCA patterns shown in Figures 1 and 3, and for various scalings of these patterns. The large blue circle "P" corresponds to the first PCA pattern. The large red circle "D" corresponds to the first DCA pattern. The blue line corresponds to possible scalings of the first PCA pattern. The red line corresponds to possible scalings of the first DCA pattern. The interesting properties of DCA arise from the fact that the red line is above the blue line. The small red circle labelled "1" corresponds to a scaling of the first DCA pattern that reduces the rainfall anomaly to the same level as that of the first PCA pattern. The small red circle labelled "2" corresponds to a scaling of the first DCA pattern that reduces the Mahalanobis distance to the same as that of the first PCA pattern. The small red circle labelled "3" is an example of a scaling of the first DCA pattern in between the scalings used for 1 and 2. The pattern corresponding to point 3 has both a larger rainfall amount and a lower Mahalanobis distance (and hence higher likelihood) than the first PCA pattern.

If we scale the first DCA spatial pattern we generate new patterns that lie on the red line in Figure 4. The slope of this line equals the ratio of total rainfall anomaly to Mahalanobis distance for the DCA pattern. The beneficial properties that DCA possesses with respect to identifying patterns with extreme totals manifest in the fact that the red DCA line is higher than the blue PCA line, and hence captures higher rainfall for a given Mahalanobis distance (or given likelihood), and, conversely, lower Mahalanobis distance (or higher likelihood) for a given rainfall. The red line is higher because the slope of this line (the ratio of total rainfall anomaly to the Mahalanobis distance) is higher for the DCA pattern. No patterns at all correspond to points above the DCA line, since this line defines the maximum rainfall for a given Mahalanobis distance (or given likelihood), by definition. Even completely uniform rainfall patterns would lie below the red line because they are very unlikely, and so would have a large Mahalanobis distance. The red line defines the "frontier" for possible total rainfall anomaly amounts.

The three small red circles on the DCA line illustrate three possible scalings of the first DCA pattern. Point 1 represents a DCA pattern scaled to be at the same horizontal level as the first PCA pattern in the diagram i.e., to have the same total rainfall anomaly. We can see from the graph that it has a lower Mahalanobis distance and hence higher likelihood value than the first PCA pattern. Point 2 represents a DCA pattern scaled to be vertically above the first PCA pattern i.e., to have the same Mahalanobis distance and the same likelihood. We can see from the graph that it has a greater total rainfall anomaly than the first PCA pattern. Point 3 represents a pattern scaled to have *both* a higher likelihood *and* a greater total rainfall anomaly than the first PCA pattern. In fact it is clear from the figure that *any* point on the red line between point 1 and point 2 is further to the top left than the PCA point and will represent a pattern that has both a higher likelihood and a greater total rainfall anomaly than the first PCA spatial pattern. As a point of interest, we note that all points in the triangle between the red and blue lines defined by the vertices 1, 2 and P also correspond to spatial patterns that would have both a higher likelihood and a greater total rainfall anomaly than the first PCA spatial pattern.

We now give details of the quantitative analysis behind the scalings and the patterns corresponding to points 1, 2 and 3 in Figure 4. Readers less interested in the quantitative details of the scalings may wish to skip to Section 5.

### 4.4. Equal Total Rainfall Anomaly Scaling

Point 1 in Figure 4 is derived using *equal total rainfall anomaly scaling* i.e., scaling the DCA pattern to have the same total rainfall anomaly amount as the first PCA pattern. This scaling does, of course, change the likelihood of the pattern, as follows. Since the ratio of the spatial pattern total rainfall anomalies for PCA to DCA is $1/3.63 = 0.28$, if we scale the DCA spatial pattern by 0.28, then the scaled DCA pattern will have the same total rainfall anomaly as the PCA pattern. The Mahalanobis distance for this new scaled DCA pattern is $6.81 \times 10^{-4}$, which is 0.28 times less than before. This is now lower than the Mahalanobis distance for the unit-scaled PCA spatial pattern and hence the likelihood for the scaled DCA spatial pattern is higher than that of the PCA spatial pattern (as seen in the diagram by the fact that the point 1 is to the left of point P). In summary, by scaling the first DCA spatial pattern by 0.28 we have derived a pattern which could occur in reality, if the multivariate normal distribution is correct. The pattern has the same total rainfall anomaly as the first PCA spatial pattern, but a higher likelihood. It is therefore a better indicator of how a large total rainfall anomaly might occur in the future than the first PCA spatial pattern, because of the higher likelihood.

### 4.5. Equal Likelihood Scaling

Point 2 in Figure 4 is derived using *equal likelihood scaling*, in which we scale the DCA spatial pattern so that it has the same likelihood as the PCA spatial pattern. Since the ratio of the Mahalanobis distances for the unit vector patterns is 0.94, if we scale the DCA spatial pattern by 0.94, then the Mahalanobis distance of the scaled DCA spatial pattern will be the same as that of the PCA pattern. This also makes the likelihoods the same and puts point 2 on the same vertical line as point P. The rainfall of the DCA spatial pattern will scale to 0.94 times its original value, and becomes 45.0, which is still 3.41 times the rainfall of the PCA spatial pattern. In the diagram we see that point 2 is directly above point P. In summary by scaling the first DCA spatial pattern by 0.94 we have derived a spatial pattern which has the same likelihood as the PCA spatial pattern, but has a greater total rainfall anomaly. It is therefore a better indicator than the first PCA spatial pattern of how large total rainfall anomalies might materialize in the future, because of the higher total rainfall anomaly at the same level of likelihood.

### 4.6. Intermediate Scaling

Point 3 in Figure 4 is derived as an *intermediate scaling* in between the equal total rainfall anomaly scaling and equal likelihood scalings described above, and the results are then more interesting than the previous two scalings in terms of mathematical properties. We apply a scaling of 0.61 to the DCA spatial pattern, which is half way between the 0.28 and 0.94 scalings used above, and we thus achieve a total rainfall anomaly of 29.1 (2.2 times the rainfall of the PCA spatial pattern) and a Mahalanobis distance of $1.5 \times 10^{-3}$ (0.65 times the Mahalanobis distance of the PCA spatial pattern). We see that by scaling the first DCA spatial pattern in this way we have created a spatial pattern which *both* has a greater total rainfall anomaly *and* has a higher likelihood (has a lower Mahalanobis distance) than the first PCA spatial pattern. In fact any scaling of the first DCA spatial pattern between the two scalings used above, and so any scaling in the range (0.28, 0.94), would have this property. Since the pattern we have created has a higher likelihood and a greater total rainfall anomaly than the first PCA spatial pattern it is presumably more relevant for understanding extreme floods and droughts.

In general it is not necessary to actually calculate these scalings, as we have done here. It is just sufficient to know that they exist, and the existence of such an intermediate scaling with the these properties in all non-degenerate cases is proven in S8. The existence of these scalings then provides mathematical justification for using the spatial structure in the DCA pattern, not that in the

PCA pattern, as the most obvious first spatial structure to consider when studying extremes of total rainfall anomaly.

### 4.7. Equal Total Rainfall Scaling at Larger Amplitude

For the scalings discussed above in Sections 4.4–4.6 the ratio of actual likelihoods is very close to one, because the amplitudes of the patterns are very small. Both the original unscaled patterns, and the scaled patterns discussed, are very close to the mean of the distribution, where the likelihood is almost flat as a function of the spatial pattern. To create patterns with larger differences in likelihood, we have to scale both patterns to much larger amplitudes. For instance, if we start with the patterns that resulted from the intermediate scaling described in Section 4.6 above we can then apply an additional scaling to *both* patterns so that the Mahalanobis distance of the PCA pattern is 1. This is a way to achieve a reasonable amplitude for the patterns. A Mahalanobis distance of 1 is the multivariate equivalent of being one standard deviation from zero (and hence of typical amplitude). This scaling can be accomplished by scaling both intermediate patterns derived in Section 4.6 by the inverse of the Mahalanobis distance of the PCA pattern, giving a scaling of 431. In terms of Figure 4, this scaling corresponds to points on the two lines that are very far beyond the top right corner of the diagram. The rainfall anomaly totals for the PCA and DCA spatial patterns increase to 5701 and 12,554. The ratio of rainfall for the two patterns stays the same, at 2.2. The Mahalanobis distances are 1 and 0.65, and the ratio of likelihoods for the two patterns is then 1.79. In this case we have created scaled PCA and DCA spatial patterns that can both be considered as possible samples from the modelled rainfall anomaly distribution. In addition both patterns now have amplitudes typical of real variability. The scaled DCA pattern has a greater rainfall anomaly than the scaled PCA pattern and has a greater likelihood by a clear margin, and hence is more relevant for understanding possible future extremes of total rainfall anomaly.

## 5. Further Examples

We now give some further examples.

### 5.1. U.S. Summer Rainfall

In Figures 5 and 6, we show the first PCA and DCA spatial patterns for U.S. summer rainfall anomalies. The PCA pattern shows near zero rainfall in the NW and a large-scale positive rainfall anomaly in the SE. The DCA pattern is similar, but shows greater positive rainfall anomalies in the NW and north central region. The explained variances of the patterns are 16.4% and 15.8% respectively. The PCA pattern suggests that there may be a recurrent pattern of rainfall anomalies on this timescale with this shape. The DCA pattern suggests that there may also be a recurrent pattern with almost the same explained variance that includes larger same-sign rainfall anomalies in the NW and the north central region. This is interesting to know, from the point of view of understanding possible floods or droughts on national spatial-scales, and is not a conclusion that could be drawn from the PCA pattern alone.

The ratios of total rainfall anomaly to Mahalanobis distance for the PCA and DCA patterns are 16,723 and 18,342, respectively. The ratio of these ratios (DCA to PCA) is 1.1. Relative to the winter example, where the ratio of these ratios was 3.41, this summer ratio is relatively close to one. The PCA and DCA patterns are more similar in terms of total rainfall, and the extra insight gained from looking at both patterns is somewhat less dramatic than it was for the winter rainfall.

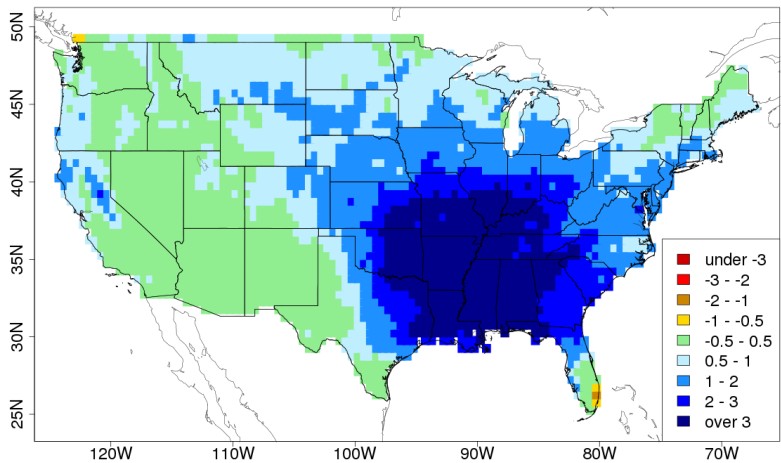

**Figure 5.** The first PCA spatial pattern for U.S. summer rainfall anomalies. This pattern maximises explained variance.

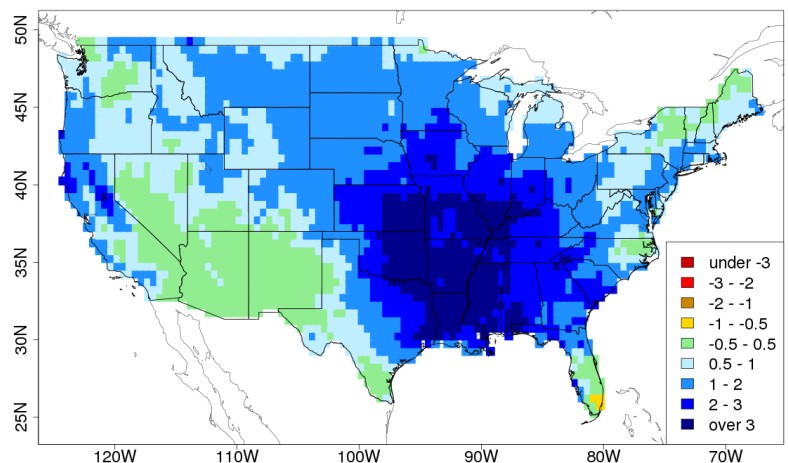

**Figure 6.** The first DCA spatial pattern for U.S. summer rainfall anomalies. With appropriate scaling this pattern is both more likely and represents a greater total rainfall anomaly than any given scaling of the first PCA spatial pattern. As a result it is more appropriate for understanding possible extremes in total rainfall anomalies than the first PCA spatial pattern.

### 5.2. China Winter Rainfall

Figure 7, panels (a) and (b), show the first PCA and first DCA spatial patterns respectively for China winter rainfall. The patterns are concentrated in the south and east of the country: much of the rest of the region is very dry climatologically. The explained variances of the patterns are 26.6% and 25.5% respectively. The PCA and DCA patterns are very similar, with the main difference being that the DCA pattern extends slightly further north. The ratios of rainfall to Mahalanobis distance are close: 7627 for PCA and 8800 for DCA. Because the patterns and the ratios are so similar, we conclude that this single pattern shape is effectively both the pattern that maximises variance, and the pattern that maximises rainfall. The situation is therefore less complex than we saw in the U.S.

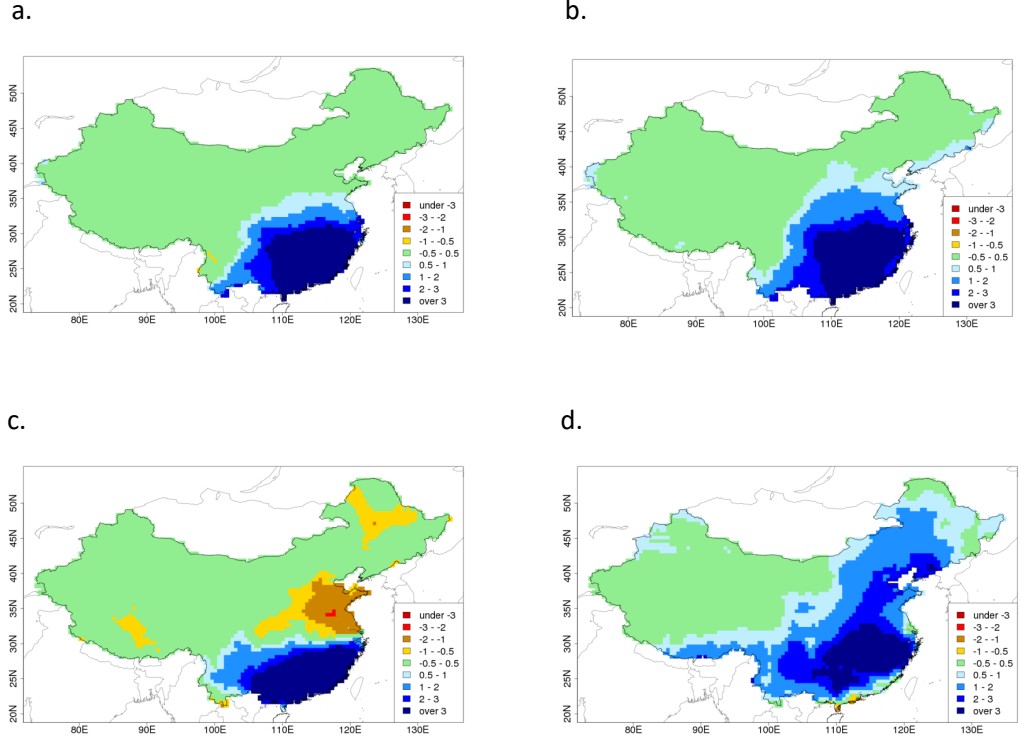

**Figure 7.** PCA and DCA first spatial patterns for winter and summer rainfall anomalies in China. Panels (**a,b**) show winter PCA and DCA respectively. Panels (**c,d**) show summer PCA and DCA respectively.

*5.3. China Summer Rainfall*

Figure 7, panels (c) and (d), show the first PCA and first DCA spatial patterns respectively for China summer rainfall. The explained variances of the patterns are 15.6% and 13.1% respectively. The situation is more complex than for winter. The first PCA pattern suggests that there is a recurring pattern of climate variability that drives a north-south rainfall dipole in eastern China. The first DCA pattern suggests that there is also a differently shaped recurring pattern of climate variability with almost the same explained variance that creates same-sign rainfall anomalies over almost the whole of eastern China and some parts of central southern China. This latter pattern would seem to be more relevant for understanding the largest scale droughts and floods in the region. The ratios of rainfall to Mahalanobis distance for the two patterns are 6090 for the PCA pattern and 14540 for the DCA pattern. The DCA pattern therefore captures 2.4 times as much rainfall overall as the PCA pattern, for any given likelihood, further emphasizing its relevance for understanding large-scale floods and droughts.

## 6. Discussion

We have described a method for pattern identification in spatially correlated multivariate space-time datasets, that we call Directional Component Analysis (DCA). For spatially multivariate normal data the method finds patterns with the highest likelihood subject to a linear constraint. For non-normal data the method can be described as finding the patterns with the highest Mahalanobis consistency subject to the same constraint. In the examples we have presented we used rainfall anomaly data, and a sum-of-all-data-points constraint that represents total rainfall anomaly, in order to find the rainfall anomaly patterns with the highest likelihood for any given level of total rainfall anomaly. The patterns are a reasonable answer to the question: what single spatial pattern is most likely to drive

large total rainfall anomalies in the future? Applying the method to U.S. winter rainfall anomalies we were able to derive a pattern of rainfall that has both a higher likelihood and a significantly greater total rainfall anomaly than the first PCA spatial pattern while having nearly the same explained variance. The first PCA spatial pattern contains large regions of positive and negative rainfall, while the DCA spatial pattern is more uniform. Based on its mathematical properties this first DCA spatial pattern is arguably the single pattern which has the greatest relevance for understanding future floods and droughts at this spatial and temporal scale (at least within the limitations of linear analysis). The actual patterns seen in the example support that very clearly: the more uniform rainfall in the DCA pattern is more likely to drive national-scale floods or droughts than the combination of positive and negative anomalies in the PCA pattern. We have also considered results for U.S. summer rainfall anomalies, and winter and summer rainfall anomalies for China, which has yielded similar insights.

The original motivating project for this work involved designing a simple methodology to identify representative extreme flood and drought scenarios in various parts of the world, for use in risk management, and the DCA spatial patterns seem like they may form part of a solution to this problem. For instance, the first DCA spatial pattern could be used as part of a methodology for simple quantification of extreme flood and drought scenarios via the following steps:

- A target spatial domain and timescale needs to be identified (in our example: the continental U.S. for a seasonal timescale)
- A target return period would be identified (such as 200 years return period)
- Standard methodologies from extreme value theory could be used to estimate the total rainfall anomaly or total drought index over the domain at that return period.
- Given this total rainfall anomaly amount the first DCA spatial pattern could be scaled to give exactly that rainfall amount. It is an appropriate pattern to represent possible rainfall extremes at that return period, since it has a higher likelihood than any other pattern with that total rainfall anomaly (by definition of DCA)
- The DCA spatial pattern so derived could then be used to drive impact models

One could imagine similar applications for deriving patterns of extreme temperature for understanding heatwaves. In both cases (rainfall or temperature) we emphasize that the use of single forcing patterns is an extreme simplification, relative to more complex models of flood, drought or heatwaves that are based on the full distribution of possible outcomes, not just a single pattern. However, simple models can play a useful role when time and resources are limited, or when only very approximate or initial answers are required.

One limitation of basic DCA patterns in this context is that, like PCA patterns, they are linear in that they are based entirely on the covariance matrix of the data, and do not account for correlations that change with the intensity of rainfall. To account for that effect, a more detailed analysis would be necessary. That might consist of deriving PCA or DCA patterns based on data censored to only include more extreme values, for instance.

DCA patterns also shed interesting light on the use of truncated PCA for simulating surrogate data, in which the first $n$ PCA patterns are retained, and the remaining patterns are discarded and replaced by a simple noise model such as white or red noise [8]. If the first DCA pattern projects onto the discarded PCA patterns, then simulated data from truncated PCA will fail to capture that pattern. In other words, in the context of rainfall simulation, the simulated data may not capture the pattern that maximises total rainfall anomaly at a given likelihood. This may be unfortunate if the simulation of extreme scenarios with large total rainfall anomaly is important. To avoid this one could consider basing simulation on a truncation of the series of DCA patterns instead, as follows. First, the data $X$ would be decomposed using DCA into:

$$X = GLQ^T \tag{21}$$

where the matrix $G$ contains the DCA spatial patterns, the matrix $L$ is diagonal, and the matrix $Q$ contains time series for each pattern. The time series in $Q$ may be correlated. To model $Q$ one might therefore use PCA, giving:

$$Q^T = E\Lambda P^T \tag{22}$$

where the new time series $P$ are now uncorrelated and easy to replace with simulated values. Combining these expressions gives:

$$X = GLE\Lambda P^T \tag{23}$$

Truncation can then be applied via the DCA patterns, by retaining just the first $m$ columns of $G$. Truncating using DCA in this way will ensure better retention of patterns with large total rainfall anomaly amounts than truncation using PCA. On the other hand, it will lead to the retention of less total explained variance as compared to truncated PCA with the same level of truncation. Which is to be preferred depends on the application.

There are various possible extensions of this research. One would be to consider other directions for the direction vector $r$ than uniform rainfall anomaly. An obvious choice would be to use $r$ to weight the different grid points so as to reflect different levels of possible impacts at different locations. For instance, when considering extreme wind one might want to use $r$ to weight populated areas more heavily than unpopulated areas.

One could also mix concepts from PCA and DCA. Using two Lagrange multipliers, it is possible to derive patterns that maximise likelihood subject to both a linear and a normalisation constraint, using the Lagrange function:

$$c = -g^T C^{-1} g + \lambda_1 (g^T r - 1) - \lambda_2 (g^T g - 1) \tag{24}$$

The solutions to this equation lie in-between the PCA and DCA patterns (in some sense), depending on the values of the Lagrange multipliers.

It is also possible to consider DCA but with non-linear constraints on the unknown pattern. For instance, with a quadratic impact function of the form $g^T M g$, where $M$ is a matrix, we have:

$$c = -g^T C^{-1} g + \lambda (g^T M g - 1) \tag{25}$$

the solutions of which are the eigenvectors of the matrix product $CM$.

One could also consider using a cost function of the general form:

$$c = -M^2 + \lambda (f(g) - 1) \tag{26}$$

Non-linear constraints for $f(g)$ may make sense in applications where impact is a nonlinear function of the variable, as it often is. If $f(g)$ is then approximated using $f(g) \approx r^T g + g^T M g$ we have

$$c = -M^2 + \lambda (r^T g + g^T M g - 1) \tag{27}$$

which has the solutions $g \propto (I - \lambda CM)^{-1} \lambda Cr$.

There are also various other potential extensions and applications of DCA. It would be possible to consider applying some of the variations and extensions used for PCA, such as application to correlation matrices rather than covariance matrices, to DCA patterns. One could investigate the weather and climate patterns associated with the first DCA pattern by regression of other weather and climate fields, such as mean sea level pressure, onto the time series $T$. In addition, one could compare the first DCA spatial pattern between observations and numerical model output as a way of evaluating how well the numerical model captures spatial extremes.

Finally we note that physical interpretation of both PCA and DCA patterns is sometimes difficult. Both are defined purely in terms of the observed statistics, and do not take any physics into account

directly. If the statistical assumptions are incorrect then they may represent patterns that could not occur in nature.

**Supplementary Materials:** The following are available online at http://www.mdpi.com/2073-4433/11/4/354/s1.

**Funding:** This research received no external funding.

**Acknowledgments:** Many thanks in particular to Farid Ait-Chaalal who performed the numerical calculations, created the maps, discussed the content and reviewed an early version of the text. Also thanks to Stephen Cusack who wrote the original computer code, and to the various other people I have discussed this work with, including Enrica Bellone, Arno Hilberts, Jo Kaczmarska and Christos Mitas. Also thanks to the anonymous reviewers whose suggestions greatly improved the final version.

**Conflicts of Interest:** The author works as a consultant to RMS Ltd, a company that builds mathematical models of extreme weather events and their impacts. The results of this particular piece of research do not currently form part of any RMS product, nor are there currently any plans to use them in any RMS product.

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
