# Peer review of "An Alternative to PCA for Estimating Dominant Patterns of Climate Variability and Extremes, with Application to U.S. and China Seasonal Rainfall"

_atmosphere, doi:10.3390/atmos11040354_

Round 1

Reviewer 1 Report

In this paper the author introduces an alternative pattern identification algorithm (DCA), similar to PCA, that can be used to explicitly optimize for spatial extremes. The analysis is thorough and useful, but I recommend to revise some major points. More specifically:

1) As the approach of this subject is rather mathematical, it would be useful for the reader to find more clear explanations of the method and the procedure (especially in the example) throughout the text. I suggest that the author make the appropriate changes to make the text more understandable for all possible readers

2) In the example the author should provide some diagrams to illustrate the described differences between PCA and DCA in each case, as it would make it more clear and easy to understand

Author Response

I would like to thank the reviewer for taking the time to read the paper and make thoughtful and helpful comments.

I have made major revisions to the paper, based on this review, which I hope will improve the paper significantly.

Point 1): more clear explanations needed

I have made a large number of changes to the text of the article, in all sections, with a focus on plain English, to make the material easier to read.

There are too many changes to list individually, but just to give some examples:

  1. I have clarified the goals of the research in the abstract and in the introduction
  2. I have moved the last part of the introduction into sections 2 and 3, which makes much more logical sense
  3. I have explained the use of Lagrange multipliers in plain English
  4. I have explained the use of the multivariate normal distribution in plain English
  5. I have explained the mathematics in the derivations in plain English
  6. I have added a better plain English explanation of the DCA solution and how the equations that define the solution can be used
  7. I have explained much of the terminology (e.g. Mahalanobis distance) in plain English
  8. I have divided section 3 into subsections to make it easier to follow
  9. I have reworded some parts of the discussion

Hopefully these changes will go some way to addressing this concern.

Point 2) (that the example was difficult to follow):

I agree that the example section was difficult to follow. The suggestion to add a diagram to help explain the example was an extremely good one. I have created a new diagram (figure 4) that I believe clarifies very well the differences between PCA and DCA, and also makes the explanation of the whole issue of scaling of patterns 100x easier to understand by making it graphical. It has certainly helped my own understanding. In addition I have over-hauled the example section by:

    1. Changing the example to winter only, which makes more sense than year round from a climatological point of view
    2. Adding a better explanation of how the example illustrates clearly the benefit of DCA over PCA
    3. Adding a new section which explains scaling in a simple graphical way using the new diagram
    4. adding 3 extra examples (US summer rainfall, and China winter and summer rainfall), to illustrate the method in different situations. The results are interesting in all 4 cases, in different ways.

I am hoping that together these changes and additions will make the example(s) much easier to follow.

Best regards,

Steve Jewson

Reviewer 2 Report

This manuscript described a method for pattern identification in spatially correlated multivariate 485 space-time datasets, DCA. While the method is interesting for geospatial data analysis, the presentation of the paper is too mathematical, and the applications to rainfall analysis failed to demonstrate the new method's advantage over the traditional PCA. The manuscript seems to be more appropriate for a mathematical or computational mathematical journal. If the author wants to make it appropriate to a meteorological journal, a better presentation, with more examples demonstrating the advantages of the new method, is seriously needed.

Author Response

I would like to thank the reviewer for taking the time to read the paper and make thoughtful and helpful comments.

I have made major revisions to the paper, based on this review, which I hope will improve the paper significantly.

1) With respect to the comment that the presentation of the paper is too mathematical:

I have made a large number of changes to try and address this. They are too many to list individually, but include:

  1. I have clarified the goals of the research in the abstract and in the introduction
  2. I have moved the last part of the introduction into sections 2 and 3, which makes much more logical sense
  3. I have explained the use of Lagrange multipliers in plain English
  4. I have explained the use of the multivariate normal distribution in plain English
  5. I have explained the mathematics in the derivations in plain English
  6. I have added a better plain English explanation of the DCA solution and how the equations that define the solution can be used
  7. I have explained much of the terminology (e.g. Mahalanobis distance) in plain English
  8. I have divided section 3 into subsections which makes it easier to read
  9. I have reworded some parts of the discussion

Hopefully these changes will go some way to addressing this concern.

2) With respect to the comments that the example failed to demonstrate the new method’s advantage over PCA, and that there need to be more examples:

In response to this comment, I have completely redesigned the example, and added 3 new examples. In each of these 4 cases I have described why I believe that the results do demonstrate the value of the method, in particular in the first example (which is now US winter).

As I say in the article, I would argue that the main value of the method is that:

  1. It is very simple to calculate indeed: it involves creating a covariance matrix and multiplying by a vector
  2. And by doing so, creates a pattern which (if the underlying assumptions are correct) is the recurring pattern in the historical data that captures the greatest total rainfall anomalies. In 2 of the examples (US winter and China summer) that pattern is distinctly different from the first PCA pattern, with same-sign anomalies over much larger scales, and yet still has almost the same explained variance, which is surely interesting to know for anyone studying climate variability and extremes.
  3. This all shows that the first PCA spatial pattern is very definitely not the best single pattern to look at to try and understand spatial extremes. Looking at PCA and DCA together is much more useful.

In addition, there are also other potential uses for the method, as discussed in section 6, such as data compression, building simple risk models, comparing climate models with observations, etc.

I hope that these changes and additions do now better demonstrate the advantages of the method.

3) With respect to the comment about the choice of journal:

I agree that this paper is rather mathematical for a meteorological / climate journal. However, the method is 100% designed for use in meteorology and climate science, I do believe it can have useful applications in this field, and that is the whole purpose of this research. If I were to publish in a more mathematical journal I would be worried that the method would never reach a meteorological or climate science audience and would never be used by meteorologists or climate scientists (and I would have to write another paper then to try and introduce it to meteorologists).

Hopefully the efforts I have made to rewrite the explanations in plainer English will go some way to resolving this concern about choice of journal.

4) In addition to the above…

I would note that I have added a new diagram (figure 4), and corresponding section in the paper, which makes the whole explanation of the scaling of the patterns 100x easier to understand, by making it graphical. This was previously one of the hardest parts of the paper to understand, but is now hopefully much improved.

Best regards,

Steve Jewson

Reviewer 3 Report

Reviewer Comments on Manuscript Atmosphere-729-748

General Comments:

This study presents some interesting findings of the comparisons between the widely used Principal Component Analysis (PCA) method and a new method called Directional Component Analysis (DC). The author compares PCA and DCA spatial patterns for U.S. rainfall anomalies on a 6-month timescale and finds that they are somewhat different. The paper presents some interesting results, yet the reviewer has some concerns that need to be addressed before this paper is accepted for publication.

Major Concerns:

  1. My major concern with this paper is that it’s not organized in a way that a reader could follow and understand the logic of what the author is trying to do. There are paragraphs that feel out of order and it’s difficult to understand the steps that were taken. I think the PCA and DCA sections can be greatly synthesized in a way that the author could first shortly discuss the use of PCA, argue about its advantages and limitations, and then move to the newly developed DCA method. I actually started to understand more clearly what the author was trying to do after reading the discussion section of the paper. I would like the author to revise sections 2 and 3 of the paper and organize the ideas and concepts in a way that research questions and objectives are more understandable early on and not towards the end of the paper.

Other Comments and Suggestions

  1. Paragraph 4 (lines 75-88) of the Introduction section seemed like the should be located in the respective PCA and DCA questions. I also encourage the author to write the research question in a more explicit way in the introduction. I know that the author is comparing two different methods, but I never found the main research statement in the Introduction.
  2. The author references multiple times the results of the PCA and DCA methods in Figure 2-3 in the first pages of the manuscript, yet the figures are located on page 11. The figures need to be located in the middle of the manuscript and not towards the end. It would be good to also include a figure of the sites or the raw data that were used to produce the PCA and DCA rainfall anomaly patterns.

Author Response

I would like to thank the reviewer for taking the time to read the paper and make thoughtful and helpful comments.

I have made major revisions to the paper, based on this review, which I hope will improve the paper significantly.

1) With respect to the comments about the logical explanation in the paper, the order of paragraphs, sections 2 and 3 and the objectives:

I have made a large number of changes to try and address these. They are too many to list individually, but include:

  1. I have clarified the goals of the research in abstract and in the introduction, as suggested
  2. I have moved the last part of the introduction into sections 2 and 3, as suggested, which now makes much more logical sense
  3. I have tried to improve the logical flow in general across the first 3 sections, including dividing into more subsections, which makes these sections easier to follow
  4. I have explained the use of Lagrange multipliers in plain English
  5. I have explained the use of the multivariate normal distribution in plain English
  6. I have explained the mathematics in the derivations in plain English
  7. I have added a better plain English explanation of the DCA solution and how the equations that define the solution can be used
  8. I have explained much of the terminology (e.g. Mahalanobis distance) in plain English
  9. I have reworded some parts of the discussion
  10. I have moved some phrases from the discussion section earlier in the document

Hopefully these changes will go some way to addressing the issues raised

2) With respect to the comment about the location of the figures:

I have moved the figures earlier in the document.

3) With respect to the comment about the raw data:

The data are in fact gridded data derived from observations by Harris et al. The gridding was done by them not us. This wasn’t clearly explained before: I have now explained this more clearly.

4) In addition to the above…

I would note that I have added

  1. a new diagram (figure 4), and corresponding section in the paper, which makes the whole explanation of the scaling of the patterns 100x easier to understand, by making it graphical. This was previously one of the hardest parts of the paper to understand, but is now hopefully much improved.
  2. 3 more examples, with discussion (and I have changed the first example too and rewritten the discussion of it). This gives more chance to see the outputs and insights that can be derived from the method. In two of the examples the DCA results are significantly different from the PCA results, which hopefully emphasizes the value of method.  

Best regards,

Steve Jewson

Round 2

Reviewer 1 Report

The author has taken into consideration and addressed my suggestions and comments. The manuscript and the methods are now more clear and easy to understand. The addition of examples is very useful. I hope that this will be a rather beneficial method for researchers.

Reviewer 2 Report

The authors have addressed all my comments, and I hope other colleagues in atmospheric research community can benefit from the new statistical method.

Reviewer 3 Report

I want to thank the author for taking into consideration my comments and for reviewing the manuscript. The manuscript has been greatly improved and now I feel comfortable in recommending its acceptance for publication.